# Urethral Injuries: Diagnostic and Management Strategies for Critical Care and Trauma Clinicians

**DOI:** 10.3390/jcm12041495

**Published:** 2023-02-13

**Authors:** Anish B. Patel, E. Charles Osterberg, Praveen N. Satarasinghe, Jessica L. Wenzel, Sabah T. Akbani, Saad L. Sahi, Brent J. Emigh, J. Stuart Wolf, Carlos V. R. Brown

**Affiliations:** Department of Surgery and Perioperative Care, The University of Texas at Austin Dell Seton Medical Center, Austin, TX 78701, USA

**Keywords:** urethral trauma, urethral injury, penetrating urethral injury, blunt urethral trauma, iatrogenic urethral injury

## Abstract

Urologic trauma is a well-known cause of urethral injury with a range of management recommendations. Retrograde urethrogram remains the preferred initial diagnostic modality to evaluate a suspected urethral injury. The management thereafter varies based on mechanism of injury. Iatrogenic urethral injury is often caused by traumatic catheterization and is best managed by an attempted catheterization performed by an experienced clinician or suprapubic catheter to maximize urinary drainage. Penetrating trauma, most commonly associated with gunshot wounds, can cause either an anterior and/or posterior urethral injury and is best treated with early operative repair. Blunt trauma, most commonly associated with straddle injuries and pelvic fractures, can be treated with either early primary endoscopic realignment or delayed urethroplasty after suprapubic cystostomy. With any of the above injury patterns and treatment options, a well thought out and regimented follow-up with a urologist is of utmost importance for accurate assessment of outcomes and appropriate management of complications.

## 1. Background

Urethral trauma is defined as forceful damage to the urethra [1,2] and accounts for roughly 4% of all genitourinary trauma [1,2]. Urethral disruption occurs in approximately 10% of blunt pelvic trauma injuries, and in 35–40% of penetrating injuries to the penis [3,4]. Approximately 65% of both blunt and penetrating traumatic urethral injuries result in a complete urethral disruption, while the remaining 35% result in at least a partial urethral tear [1]. Urethral injuries can also have a series of devastating long-term consequences, such as complications of impotence, stricture formation, and incontinence.

Any type of trauma can affect the urethra; however, specific injury patterns affect the urethra at higher incidence. The majority of urethral trauma is caused by blunt injury, penetrating injury, and/or iatrogenic injury. Most major urethral injuries are due to blunt trauma caused by mechanisms such as motor vehicle collisions, kicks, or falls. Blunt trauma accounts for up to 90% of urethral injuries [5]. Penetrating trauma accounts for 9–10% of urethral injury and results primarily from mechanisms such as gunshot wounds, stabbing incidents, or insertion of foreign objects [4]. Lastly, iatrogenic urethral trauma accounts for less than 1% of urethral injury and is often the result of difficult urethral catheterization or transurethral procedures such as prostate or tumor resection that lead to stricture formation [4].

Often times, the anatomy of the urethra determines the pathology of urethral injury. For a better understanding of urethral trauma, it is necessary to subdivide the urethra into the anterior and posterior anatomical divisions (Figure 1). The anterior urethra runs 15 centimeters in length and is distal to the perineal membrane, while the posterior urethra runs 3 centimeters in length and is proximal to the perineal membrane [6]. The anterior urethra is further broken down into the bulbar, penile urethra, fossa navicularis, and urethral meatus from proximal to distal while the posterior urethra includes the bladder neck, prostatic urethra, and membranous urethra from proximal to distal [6].

Most anterior urethral injuries (AUI) are caused by self-instrumentation, penetrating injuries, straddle injuries, crush injuries, iatrogenic trauma, and sexual misadventure [7]. Because the anterior urethra is composed of superficial epithelium that lies directly on the spongy, vascular erectile tissue of the corpus spongiosum, trauma can cause the epithelium to be easily breached and result in adverse effects such as hematuria and erectile dysfunction [7]. In contrast, most posterior urethral injuries (PUI) are caused by pelvic fracture and blunt trauma from massive deceleration events or pelvic crush injuries [7]. Rather than compromising the spongy epithelium, PUI is caused by blunt compressive force that results in pelvic ring disruption [8]. These injuries are often referred to as pelvic fracture urethral injuries (PFUI) as they are most commonly associated with a pelvic fracture [8]. Pelvic fractures occur in 10% of blunt trauma cases, and reportedly 5–25% of pelvic fractures are associated with a PUI [8,9].

With an abundance of literature on urethral trauma, the goal of this review is to provide clinicians a clear, concise summary of urethral trauma management by highlighting etiology, diagnosis, and management of blunt, penetrating, and iatrogenic trauma of the anterior and posterior urethra.

## 2. Methods

The authors reviewed the literature surrounding the different types of urethral trauma dating from 1980–2018. The PubMed search terms included the following: urethral trauma, urethral injury, iatrogenic urethral trauma, foley trauma, urethra injury, penetrating urethral trauma, penetrating urethral injury, blunt urethral trauma, blunt urethra trauma, blunt urethral injury, posterior urethral injury, pelvic fracture urethral injury, posterior urethral trauma, anterior urethral injury, and anterior urethral trauma. We included both the European Association of Urology (EAU) urotrauma guidelines and American Urological Association (AUA) urotrauma guidelines [1,2]. Primary and review articles were included in our search, while all level 5 evidence was excluded. Overall, we found 80 articles that met inclusion criteria and that were utilized in our review. The literature was categorized into three major types of urethral injury: (1) iatrogenic, (2) penetrating, and (3) blunt. The three major types of urethral injury were further stratified to investigate the effects of each type of urethral injury on the anterior and posterior urethra. We collected the incidence, etiology, diagnostic work-up, and management for each of the three types of urethral injury.

## 3. Discussion

### 3.1. Overall Consensus on Diagnosis

A thorough clinical exam is essential for making a diagnosis of urethral injury. The most characteristic clinical sign of urethral injury in pelvic fractures is fresh or dry blood at the urethral meatus [3]. In fact, blood at the meatus is seen in 37 to 93% of PUI [3,4]. However, the amount of blood present does not indicate the extent of the injury [5]. The presence of a high-riding prostate has relatively low sensitivity in determining if a PFUI is present [3].

Before advances in urethrography, blind catheterization was the primary diagnostic modality for suspected urethral injury [6,7]. A theory that a partial urethral injury could be exacerbated by urethral catheterization and mature into a full urethral injury has remained to be dogma [8]. Current guidelines recommend a single, gentle attempt at urethral catheterization by a urologist with blood per meatus on exam following retrograde urethrogram (RUG) [2,9].

The EAU and AUA guidelines state that RUG is the gold standard diagnostic modality in all cases of suspected urethral injury [1,2]. RUG is operator-dependent and is recommended to be performed by the treating urologist, as a disparity has been found to exist between independently reported urethrography compared to the urethrography interpreted by the urologist treating the patient [10,11]. RUG is performed by first properly positioning the patient confirmed with a scout film in order to allow clear visualization of the urethra. The procedure is done with the patient supine, pelvis elevated at 30 to 45 degrees from the horizontal plane, upper thigh positioned straight, and the thigh nearest to the table flexed at 90 degrees. One technique for urethrography involves inserting a 14 Fr Foley catheter into the fossa navicularis and inflating the balloon to 2 cc, which is then connected to a 60 cc catheter syringe with undiluted, water-soluble, full-strength contrast material. Finally, 30 cc of contrast material is injected into the urethra and a radiograph is taken once at the end of the injection. With this technique it is easy to visualize any extravasated contrast material or urethral stenosis/stricture with failure of dye to enter the bladder; therefore, fluoroscopic imaging is not necessary (Figure 2) [6].

RUG is accurate in fast-paced trauma settings and can be performed in a timely manner [6]. Ultimately, the goal is to detect contrast medium extravasation around the corpus spongiosum and to determine if there is a continuous, laminar contour of flow [9]. Furthermore, a RUG can be used to stage urethral injuries. Urethral injuries may be classified as Types 1–5 according to location and extent of the trauma; however, the distinctions of each are beyond the scope of this review [12]. RUG is not accurate at determining the severity of urethral trauma with regards to density of scar tissue, prostatic displacement, avulsion of the corpus cavernosum, and surrounding soft tissue injury [8,13]. In addition, long urethral defect lengths are underestimated due to improper filling of contrast medium [13]. Nonetheless, the threshold for performing an RUG should be low in any male patient who presents with blood at the meatus, perineal or penile hematoma, those who cannot void or have difficulty voiding, or a concomitant saddle-type fracture [14]. Although trauma can lead to significant urethral disruptions in men, female urethral injury is rare due to the shorter length, higher elasticity, and relative mobility of the female urethra [15]. However, female urethral injuries can be associated with up to 6% of pelvic fractures [5,6,14,16]. For these reasons, in female patients with suspected urethral injury and/or hematuria, labial swelling, or blood within the vaginal introitus, they should undergo urethroscopy to more effectively assess for bladder and urethral injuries [1].

Each millimeter of pubic symphysis widening is associated with an approximately 10% increased risk of urethral injury [17]. Interestingly, Luckhoff et al. reports that male patients suffering pelvic fractures with pubic symphysis disruption on initial anteroposterior X-ray and/or clinical signs of urethral trauma had a sensitivity of 100% for urethral injury, suggesting that RUGs are not necessary in patients without disruption of their pubic symphysis or no clinical signs of urethral trauma [8].

Although the EAU suggests there is no role for computed tomography (CT) or magnetic resonance imaging (MRI) in the evaluation of urethral injuries [1], CT scans can help delineate whether there is a concomitant organ injury [18]. In circumstances where RUG is not readily available or in particularly busy trauma settings, CT scan can be a useful proxy to discover patients that are at high risk of urethral injury [19]. Such CT scan findings that are associated with urethral injuries include distortion or obscuration of the urogenital diaphragm fat plane, prostatic contour, and/or bulbocavernosus, and the presence of a hematoma in the ischiocavernosus and/or obturator internus muscles [19]. Although CT scan can help specify the type of urethral injuries, it should not be used to replace RUG as a diagnostic modality [19]. With regards to MRI, Koraitim et al. reports that in 86% of their patients with PUI, the length of urethral defect and density of scar tissue could be determined more accurately by MRI compared to RUG [13]. Important factors to consider when deciding on an appropriate treatment modality involves having an idea on the length of the urethral defect, type of prostatic displacement, and presence and severity of scar tissue [13] and thus, MRI may be a useful preoperative diagnostic tool.

Lastly, for suspected iatrogenic urethral injuries that occur during urological or gynecological procedures, cystoscopy remains the gold standard for diagnostic evaluation [16]. Cystoscopy can be performed via both retrograde and/or anterograde routes through an established suprapubic tract into the bladder to diagnose a urethral injury [1].

Sonourethrography is able to accurately detect soft tissue trauma and may even more accurately determine stricture length than RUG, but should not replace a RUG [20]. Considering the two-dimensional nature of RUG, penis traction and position of the patient will alter the appearance of the stricture and thus alter the accuracy of the stricture length [20,21]. Sonourethrography can safely and in a timely manner evaluate urethral caliber, the presence of periurethral spongiofibrosis, as well as other intraurethral lesions more accurately than RUG; however, its role in a trauma setting is limited [20,21]. Sonourethrography has a 56% sensitivity for detecting spongiofibrosis and up to 83% sensitivity in detecting high grade severity [21]. As with all ultrasonographic studies, interpretation of the results is subject to observer bias [21].

### 3.2. Iatrogenic Urethral Injuries

#### 3.2.1. Incidence

Iatrogenic urethral injuries resulting from trauma during catheter placement occur in approximately 0.3% to 25% of inpatient hospital visits [22]. Urethral stricture as a result of iatrogenic urethral injury can form in up to 17% of patients following radical prostatectomy [23]. Furthermore, urethral stenosis results in 2% to 26% of patients undergoing prostate cancer radiotherapy [24].

#### 3.2.2. Mechanism

Iatrogenic urethral injuries are defined by a spectrum of disease, ranging from minor contusions to severe strictures. They can occur in both anterior and posterior urethral segments [6]. Iatrogenic urethral injuries can be secondary to a variety of causes, of which traumatic urethral catheter insertion and/or prolonged placement are the most common causes [16]. This is likely due to forced catheterization across the posterior urethra and/or premature inflation of the balloon within the urethra [22]. In addition, many injuries occur from traumatic transurethral procedures and other forms of urethral instrumentation, which includes catheterization, external beam radiation, and a variety of transurethral procedures [14,16,23,25]. Rarer causes of iatrogenic urethral trauma include surgical removal of penile warts [26], extracorporeal circulation during cardiac revascularization surgery [27], and bladder-drainage before pancreatic transplantation in diabetic patients [28]. Female urethral injuries are rare, and most iatrogenic injuries follow instrumentation use during obstetrical and gynecological procedures, with a smaller proportion occurring after urological sling procedures [29].

#### 3.2.3. Management

For prevention of traumatic urethral catheterizations, appropriate education surrounding catheterization is recommended for all healthcare providers that routinely place catheters [22,30,31]. Specifically, education should be centered around basic urological anatomy and safe techniques. Prior to catheter insertion, s-shaped urethra should be straightened without compressing it. Catheters should be inserted to the hub and the provider should wait for urine to return before inflating the balloon with sterile water or air. In clinical settings that have the capacity, direct vision catheters can be utilized to allow for better visualization of lower urinary tract anatomy prior to catheterization to prevent unsafe blind passages [32]. Further preventative measures include intermittent catheterization and utilization of less irritative catheters [6]. Less irritative catheters are typically made of silicone-based material in contrast to materials that are more friction- and encrustation-prone, such as latex rubber or plastic [33].

In the event of repeated failed attempts of Foley catheter insertion, the initial step includes obtaining a thorough genitourinary history with specific attention to prior prostate surgery/prostatectomy, radiotherapy, or any other events leading to potentially disrupted anatomy [34]. Next, any history of benign prostatic hyperplasia should lead to the assumption that a steep angle is needed to enter into the bladder. The final step involves acting according to these assumptions. For example, for a presumed stricture, place a 12 Fr silicone, and for a presumed underlying benign prostatic hyperplasia, place an 18 Fr Coudé [34,35]. If the Foley catheter fails to insert after completion of these steps, Urology should be consulted to evaluate with bedside cystoscopy and attempt wire-guided catheterization as needed.

Once a stricture has formed, it can be surgically managed with urethrotomies, which are an endoscopic opening of the scar internally, or repeated dilations [6,36,37]. In circumstances involving iatrogenic prostatic urethral strictures, endoscopic management is preferred to indwelling catheter placement or an open procedure [37]. Lastly, urethroplasty is suggested for urethral lesions following radiotherapy [2,38,39]. In general, iatrogenic urethral injury should be managed with a simple approach focusing on early catheterization and observation followed by further intervention as needed.

Despite a wide variety of etiologies associated with iatrogenic urethral injuries, strictures continue to be the priority when caring for patients with such injuries (Table 1). Davis et al. reported several options for initial management of traumatic urethral catheterizations including percutaneous insertion of a suprapubic catheter (SPC), flexible cystoscopy plus guidewire to catheterize the bladder, transurethral catheterization performed by a urologist, catheter manipulation and re-insertion, rigid cystoscopy plus guidewire, and finally, open cystostomy plus suprapubic catheter [31]. The most common of these options remains temporary indwelling catheterization, while in difficult cases, cystoscopy and guidewire placement or SPC are alternatives [30].

### 3.3. Penetrating Urethral Injuries

#### 3.3.1. Incidence

One study reported that penetrating urogenital injuries represent 2% of all penetrating trauma injuries and 0.6% of all trauma presenting at their hospital [36]. Of these patients, 13% also had urethral injuries on presentation [36].

#### 3.3.2. Mechanism

Penetrating trauma is most often caused by motor vehicle accidents and firearms; however, it can also be a consequence of stab wounds, bites, self-mutilation, and industrial accidents [6]. Gunshot wounds remain a common source of penetrating trauma and are the etiology in 15% to 29% of urethral injuries resulting from trauma to the penis [41]. Gunshot wounds are also the most common cause of injury to the anterior and posterior urethra [16,42,43].

#### 3.3.3. Management

##### Anterior Urethral Injury

As seen in Table 2, thorough perineal or scrotal exploration with hemostasis and judicious debridement are key to managing these injuries. Careful debridement is required to minimize periurethral tissue loss [6]. Both the EAU and AUA have recommended open surgical repair unless there are other potentially fatal injuries [14]. With mild injury, conservative management techniques may be used without requiring catheterization. However, if there is severe injury, an SPC with delayed primary repair may be warranted [14]. SPC manages both partial and complete urethral disruption. Typically, all complete disruptions result in urethral stricture which later requires a urethroplasty.

While often done for blunt PUI, primary endoscopic realignment (PER) can also be performed in some cases of penetrating AUI. In fact, PER is often the recommended management strategy for penile-fracture-associated urethral injury or penetrating injury to the anterior urethra, as it reduces the risk of deformity and strictures [45,46]. PER is a minimally invasive management option that utilizes a single flexible cystoscope and wire, or in some cases, antegrade and retrograde cystoscopes in an attempt to maintain cephalad–caudal axial urethral end alignment [45]. The Seldinger Technique is then employed to place a urethral catheter across the disruption to achieve continuity [45,47]. Recent literature indicates that PER is simple, safe, rapid, and nontraumatic, reducing urethral injury compared to the open approach [37]. Defects that are up to 2 cm in size within the bulbar urethra and up to 1.5 cm in size within the penile urethra may be repaired with primary direct anastomosis; however, repair of larger defects should be deferred until a later date to allow for proper planning of tissue transfers [48]. In the meantime, these patients require placement of an SPC [46].

##### Posterior Urethral Injury

The first step in management of a penetrating injury to the posterior urethra is to obtain urinary diversion with either indwelling catheterization or SPC [49,50]. Once urine is diverted, operative repair should be tailored to the location of the injury. If in the bladder neck, then the ureters must be evaluated and passage of a ureteral stent for staging is warranted [49,51]. If in the prostatic urethra, the prostate must be evaluated for viability and to determine whether a prostatectomy is necessary [51]. If in the membranous urethra, then a urethroplasty is preferred once the patient is stable, which can be performed by either perineal or abdominoperineal approaches. Regardless of location, a penetrating PUI should be repaired as early as possible if there is any concern for potential fistula formation between the urethra and rectum or urethra and vagina [51].

Therefore, the cornerstones of management of a penetrating PUI are urinary drainage, PER when possible, and a delay in definitive urethral reconstruction, if needed, until stabilization of the patient (Table 3) [43]. When the patient has stabilized and the approach to surgical reconstruction is being considered, Tausch et al. advocate for initial wide retropubic exposure via an initial lower abdominal midline incision [43]. They were also proponents of delayed perineal urethroplasty, although they used an abdominoperineal approach without pubectomy in one patient who had undergone prior perineal surgery. Of their 15 patients who underwent delayed repair, 13 (86.6%) had normal urinary flow rates and denied lower urinary tract symptoms at the time of follow-up [43]. In cases of concomitant bladder neck injury, prostatectomy should be considered to increase access for surgical correction [43]. Additionally, patients with bladder neck injury and/or complete urethral injury should be considered for primary urethral repair [42,43].

### 3.4. Blunt Urethral Injuries

#### 3.4.1. Incidence

A total of 5% to 25% of males who have a pelvic fracture will have a PFUI stemming from blunt trauma [37]. Thus, amongst patients with membranous urethral injuries related to blunt trauma, almost all have an associated pelvic fracture [6]. Furthermore, blunt injuries to the posterior urethra are four times more incident than injuries to the anterior urethra [52]. Additionally, up to 15% of patients with penile fractures have some form of urethral disruption, with partial transection being the most common injury [8,16,53]. In 20% of penile fracture cases, the anterior urethra is involved [54].

#### 3.4.2. Mechanism

Blunt trauma is most often the cause of AUI [10,42,44,55]. Injury most commonly occurs to the bulbous urethra, which, in contrast to the pendulous urethra, is fixed in space underneath the pubic symphysis [6]. Typically this results from a straddle injury in which the bulbous urethra is aggressively compressed against the pubic bone, causing significant urethral injury [6]. Additional reported mechanisms of blunt AUIs include penile fractures, motor vehicle accidents, bicycle accidents, or perineal kicks, often by bulls or horses [6,16,56]. With motor vehicle accidents, occupants are affected through massive deceleration as well as any involved pedestrian [5].

Blunt trauma resulting in PUI is often associated with complete urethral rupture, whereas blunt trauma resulting in AUI typically results in partial rupture [57]. As mentioned, the bulbous urethra is the most commonly affected segment (85%) in blunt AUIs as it is fixed underneath the pubic bone, while the bulbomembranous junction is most commonly affected in blunt PUIs [5,6,9,23]. However, the prostatomembranous urethra can also be affected primarily through a shearing force that separates the prostate from the membranous urethra, where the membranous urethra is fixed in place by the urogenital diaphragm and puboprostatic ligament [9].

Furthermore, certain pelvic fracture subtypes are associated with a higher risk for PUI, including diastasis of the sacroiliac joint and the Malgaigne fracture [53,58,59]. A Malgaigne fracture is caused by a vertical shear force leading to upward hemipelvis displacement and ipsilateral fracture of both pubic rami or pubic symphysis with associated sacroiliac dislocation or ipsilateral ilium [3,5].

With both of these types of fractures, the chances of having a resulting urethral injury increases eight-fold [53]. Conversely, the risk of urethral injury is negligible with fractures that do not involve the ischiopubic rami [16,53].

#### 3.4.3. Management

##### Anterior Urethral Injury

As with other types of urethral injury, initial urinary diversion is recommended in suspected blunt AUI [9]. While multiple attempts at blind urethral catheterization are not recommended, a trial of endoscopic urethral catheterization is acceptable [9,45]. Beyond this initial step, management is varied and contradictory at times (Table 4). The various management options include Foley catheterization with hopes of the injury healing patent across the catheter, PER, and urinary diversion with an SPC and staged urethroplasty.

Early intervention has had variable success rates, with a range of 0–81.5% success in early PER [60]. Evidence has also supported a much higher success rate seen in primary open repair when compared to PER (85.6% vs. 57.0%, respectively) [60]. A small study investigating the utility of early fluoroscopic realignment found that out of two patients with blunt AUI, one was treated successfully and one was not able to be successfully realigned [55]. Others advocate for delayed urethral repair to allow better visualization of the extent of the crush injury [9]. High success rates have been reported with a delayed perineal approach for stricture excision and primary anastomosis [9].

Park et al. reviewed data of patients presenting acutely with blunt saddle injuries that were treated with SPC diversion. These patients required less frequent subsequent procedures than those treated with initial catheterization and early PER [56]. In contrast, sixty percent (47/78) of patients had a delayed presentation with obstructive symptoms. Although these patients had longer urethral strictures than those presenting acutely, successful long-term outcomes were achieved via urethroplasty in 98% of the cases. Prior urethral manipulation was the strongest predictor of failed urethroplasty and symptom recurrence [56]. It is possible that the study by Park et al. inadvertently created a setting that showed the benefit of delayed urethral repair in blunt AUI due to a large number of patients not presenting until months after initial injury. However, due to varied results among the literature, this is an area that requires further research.


**Table 4 jcm-12-01495-t004:** Blunt Anterior Urethral Trauma.

Author	Year	Level of Study	Number of Patients in Study	Diagnostic Procedures	Management	Complications
Londergan TA, Gundersen LH, van Every MJ [55]	1997	4	6	RUG	Fluoroscopic realignment	Procedure: Requiring interval urethrotomy; prostatic abscess. ⅖ (40%) patients had impotence—consistent with other studies.
Jordan GH, Virasoro R, Eltahaway EA [9]	2006	3	NA	RUG, endoscopy	Delayed excision with primary anastomosis	Procedure: Abscess, erectile dysfunction, infection
Dobrowolski ZF, Weglarz W, Jakubik P, et al. [44]	2002	3	48	RUG, voiding cystourethrogram, endoscopy	25% surgery which included drainage, 75% conservative management	Stricture
Zhang Y, Zhang K, Fu Q [60]	2018	3	1606	RUG	Urinary diversion, primary repair (endoscopic vs. open realignment)	Infection, stricture
Park S, McAninch JW [56]	2004	3	78	RUG	Suprapubic urinary diversion, delayed presentation led to urethroplasty	Obstructive symptoms in patients with prior urethral manipulation

RUG = retrograde urethrogram.

##### Posterior Urethral Injury

In the case of pelvic fractures, immediate repositioning and stabilization is necessary to control bleeding and prevent additional damage [61]. If there is a complete urethral rupture, immediate transurethral splinting may be done using fluoroscopy or endoscopy which is previously described as PER. (Table 5) [61,62,63]. Urinary diversion through an SPC can also be performed with the notion that a delayed urethral reconstruction is necessary [6,61].

PFUI management techniques include PER of the urethra through an open or endoscopic approach, or SPC with delayed urethroplasty [5]. Early end-to-end urethroplasty has been successful in repairing the urethra in 80–90% of cases [62]. Secondary or delayed treatment is needed in unsuccessful cases or for those in which procedures could not be done within the first 10 days after injury, especially due to the presence of other life-threatening injuries [62]. Unfortunately, an SPC with delayed reconstruction after 3–6 months has been associated with high risk of urethral strictures (97%), massive blood loss ranging from 500–3150 mL, and long hospital stays (ranging from 22–28 days) [63,64]. Conversely, primary repair has a low incidence of severe urethral stricture [37]. An endoscopic approach is preferred by some due to its safety profile and less invasive nature, as open primary reconstruction can be technically challenging due to distorted anatomy and tissue which increases the risk for neurovascular damage and subsequent impotence [48].

Overall, the management of blunt urethral injury falls into one of two categories: PER versus SPC diversion and delayed repair. Either way, the initial short-term goal is to provide prompt urinary drainage with a definitive plan for management. Traditionally the most dependable and performed procedure was SPC diversion with delayed repair, but as discussed, there is a resurgence of evidence suggesting that primary realignment, whether endoscopic or open, may also be an equally appropriate option with equivocal success rates. Choosing one of these two management options remains controversial, though there are new efforts to evaluate if one approach may be more superior [65].

**Table 5 jcm-12-01495-t005:** Blunt Posterior Urethral Trauma.

Author	Year	Level of Study	Number of Patients in Study	Diagnostic Procedures	Management	Complications
Dobrowolski ZF, Weglarz W, Jakubik P, et al. [44]	2002	4	268	RUG, voiding cystourethrogram, urethroscopy, spiral CT	Cystostomy + drainage, urethral reconstruction	Procedure: Urethral stricture, urinary tract infection, bleeding, stones, fistula
Koraitim MM, Reda IS [13]	2007	4	21	Pre-operative MRI of pelvis to aid in selection of appropriate surgical approach	End-to-end urethral anastomosis	Injury: Retropubic space scarring and fibrosis, fistula, impotence
El Darawany HM [37]	2018	4	27	CT Urography, RUG	Initial Management: Suprapubic catheterization to drain bladder Endoscopic urethral realignment	Procedure: Minimal urethral stricture
Koraitim MM [66]	1996	4	100	Clinical examination Excretory urography, retrograde urethrography	Suprapubic cystostomy indicated for incomplete urethral rupture, slight urethral distraction, and critically unstable patients, and for inadequate facilities or inexperienced surgeons Primary realignment for wide separation of urethral ends or associated injury to bladder neck or rectum	Procedure: Urethral stricture after suprabpubic cystostomy Impotence and incontinence after primary realignment Impotence and incontinence is greatest after primary suturing
Velarde-Ramos L, Gomez-Illanes R, Campos-Juanatey F, et al. [3]	2016	3	NA	RUG	Primary urethral realignment, delayed urethroplasty, SPC + endoscopic realignment	Procedure: Sepsis, abscess, urinoma
Cavalcanti AG, Krambeck R, Araujo A, et al. [67]	2006	3	77	RUG, surgical exploration	Primary urethral realignment/repair	Procedure: Penile curving, corporal body injury
Goldman SM, Sandler CM, Corriere JN Jr., et al. [12]	1997	3	NA	RUG	Delayed urethroplasty	NA
Elgammal MA [68]	2009	3	53	RUG	Suprapubic cystostomy, primary urethral realignment	Procedure: Urethral stricture
Jordan GH, Virasoro R, Eltahaway EA [9]	2006	3	NA	Clinical exam: direct examination RUG, cystogram	SPC + delayed repair, primary urethral realignment, endoscopic realignment	Procedure: Compartment syndrome, rhabdomyolysis
Protzel C, Hakenberg OW [57]	2010	3	NA	Clinical exam: direct examination, digital rectal exam RUG	Suprapubic urinary diversion + antibiotics, endoscopic transurethral splinting	NA
Tezval H, Tezval M, von Klot C, et al. [18]	2007	3	NA	Clinical exam: Digital rectal exam Suprapubic cystography, RUG, CT scan	Cystostomy + antibiotics, primary urethral realignment, delayed urethroplasty	Procedure: Stricture, erectile dysfunction, distraction defects, impotence, recurrent stenosis
Rosenstein DI, Alsikafi NF [6]	2006	3	NA	Clinical exam: Digital rectal exam Suprapubic cystography, RUG	SPC, delayed urethroplasty	Procedure: Bladder rupture, vaginal laceration, rectal tears
Luckhoff C, Mitra B, Cameron PA, et al. [8]	2011	3	998	RUG, pelvic X-ray, sonography	Primary urethral realignment	Procedure: Urethral transection
Cooperberg M, McAninch JW, Alsikafi NF, et al. [69]	2007	3	134	RUG/ voiding cystourethrogram Pelvic MRI if length of stricture is in question Doppler ultrasound for patients with erectile dysfunction to determine baseline neurological or vascular compromise	Delayed Anastomotic Posterior Urethroplasty	Procedure: Scrotal hematoma, compartment syndrome (lithotomy position), decubitus ulcer, recurrent stricture

RUG = retrograde urethrogram, CT = computed tomography, MRI = magnetic resonance imaging.

##### Complications of Urethral Injuries

There are many complications that result from the different types of urethral injury. After reviewing the literature on penetrating, iatrogenic, and blunt urethral injury, several shared complications from urethral trauma surfaced.

First, when comparing AUI and PUI, there are important complication distinctions. For PUI, common complications include erectile dysfunction and incontinence (Table 3 and Table 5). For AUI, the exposure and breach of Buck’s fascia after penetration of the surface epithelium results in urine extension into the scrotum and/or abdominal wall. As a result, infection from a urinary source is possible [16]. Additionally, partial or complete tear of the anterior urethra epithelium can result in strictures and fistulas (Table 2 and Table 4).

When investigating iatrogenic, penetrating, and blunt urethral trauma, urethral injury as well as its management are both associated with risks and complications. First, different mechanisms of iatrogenic urethral injury produce different complications. Foley trauma can lead to blood at the meatus, scrotal hematoma, and eventually strictures (Table 1). In stricture disease, chronic inflammatory processes may result from repeated dilations and urethrotomies.

In cases of penetrating urethral injuries, complications may result from surgical management. Pelvic exploration and primary repair of urethral ends are associated with high rates of urinary incontinence (21%) and erectile dysfunction (56%), while delayed urethral repair following placement of an SPC is associated with lower rates of urinary incontinence (4%) and erectile dysfunction (19%) [5,66]. Management of penetrating urethral injuries associated specifically with PUI have been found to have rare complications of urethrocutaneous fistulas (2%), urinary bladder stones (3%), and inflammation of the lesser pelvis (2%) [44].

In cases of blunt urethral trauma resulting in PFUI, erectile dysfunction occurs in 20% to 84% of patients; those with persistent erectile dysfunction seem to have a neurogenic cause [68,69,70]. In patients with PFUI involving a rare injury to the pelvic plexus in which bladder and bladder neck function is compromised, sphincter weakness incontinence may result in voiding difficulty [51]. Controversially, urethroplasty may cause erectile dysfunction in patients, especially those with some degree of underlying erectile dysfunction prior to injury; however, most patients have been found to retain general sexual function despite weaker ejaculation and reduced fertility [70,71].

##### Follow-Up for Urethral Injuries

Follow-up for all urethral injuries is critical to monitor patients for complications, including stricture formation and/or recurrence, erectile dysfunction, and incontinence over the course of at least one year [2,37,66]. Stricture formation is one of the most common complications of urethral injury and repair [72,73]. Stricture recurrence is prevalent within 2 years of initial stricture treatment, but is typically discovered within weeks or months post-procedurally [74]. Therefore, follow-up visits occur at 1-, 3-, 6-, and 12-month intervals, and eventually yearly [37,40].

Visits are often comprised a combination of patient history, physical exam, uroflowmetry, retrograde urethrogram, cystoscopy, and/or post-void residual measurement [36]. Patient history should focus on discovering symptoms of scar formation such as dysuria, an increase in urination time, a feeling of incomplete emptying, straining, weak stream, as well as increased frequency and urgency [72,73,75]. RUG or cystoscopy can help delineate obstructive urinary symptoms, decrease in flow rate, and/or an increase in post-void residual volume [40]. Cooperberg et al. suggests that questions from the urinary and sexual domains of validated patient reported outcome measures such as the Expanded Prostate Cancer Index Composite can help determine patients’ reported outcomes regarding continence and potency [69]. Treatment of stricture recurrences is beyond the scope of this manuscript.

##### Areas of Future Discovery and Improvement

Amidst our exhaustive literature review, we found that the literature was lacking in the domain of penetrating PUI. We suggest developing multicenter outcomes studies that capture such rare events. Considering their relative rarity and low prevalence this is not all too surprising; however, the lack of literature might lead to an incomplete understanding, on which the trauma-astute physician cannot rely to make confident clinical decisions.

Despite reviewing a plethora of management recommendations, we were unable to find extensive literature on detailed follow-up recommendations. We suggest that clinicians develop an individualized and focused routine comprising an appropriate history, accurate physical exam, and reliably available diagnostic modalities when evaluating their patients with urethral injuries in the follow-up period, as outlined previously. Lastly, in regard to evolving surgical techniques, we believe there should be a better understanding of novel allografts and alternatives to substitute the urethral mucosa.

We believe there is potential for future studies to further investigate iatrogenic urethral injury secondary to traumatic catheterization by delineating whether catheters designed to be less irritating either in material or build can lead to a lower incidence of urethral injury. In fact, such studies are crucial to ensuring that we improve the quality of the care we provide to our patients.

## 4. Conclusions

After performing an exhaustive literature review and summarizing our findings above, we have concluded that there are a few core principles in the overall management of urethral trauma which every clinician should adopt. In diagnosing urethral injuries, RUG is the single best imaging modality and it should be the first test performed. In treating iatrogenic urethral injury, catheterization should be the standard of care with observation and intervention as needed thereafter. In treating penetrating urethral injuries, early repair is favored and every reasonable effort should be made to do so, unless the patient remains unstable or has other life-threatening injuries which take priority. Finally, in treating blunt urethral injuries initial catheterization, PER, and SPC followed by delayed urethroplasty after stricture formation are acceptable management strategies. We have concisely summarized these points in Figure 3.

## Figures and Tables

**Figure 1 jcm-12-01495-f001:**
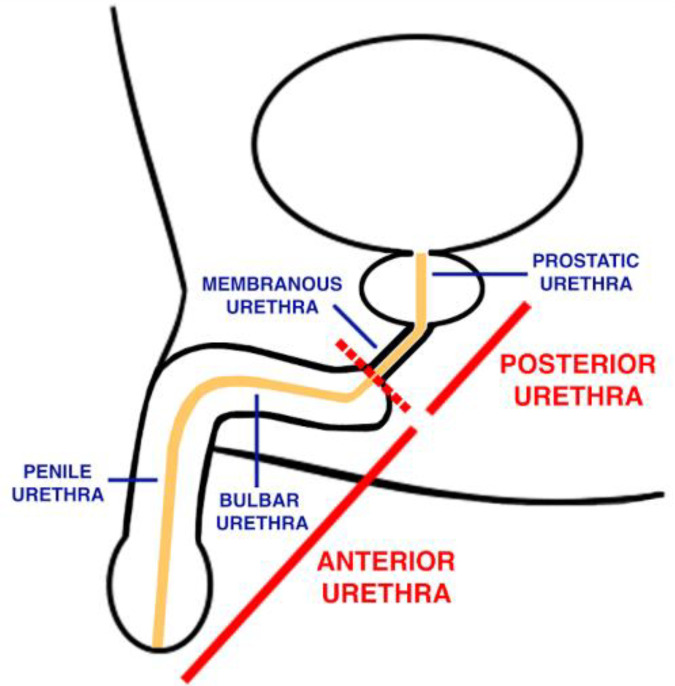
Basic urethral anatomy.

**Figure 2 jcm-12-01495-f002:**
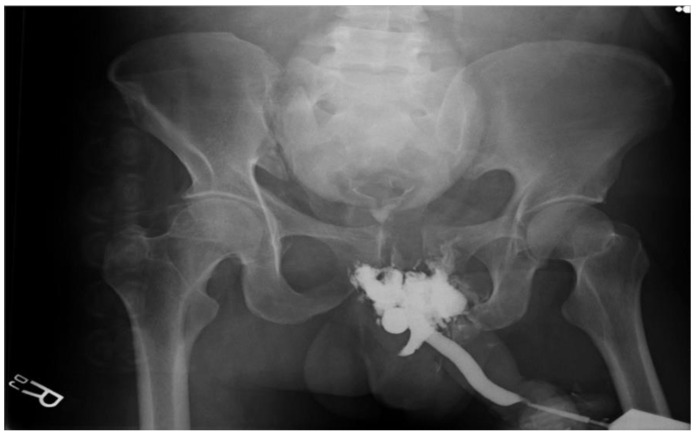
RUG indicating urethral injury.

**Figure 3 jcm-12-01495-f003:**
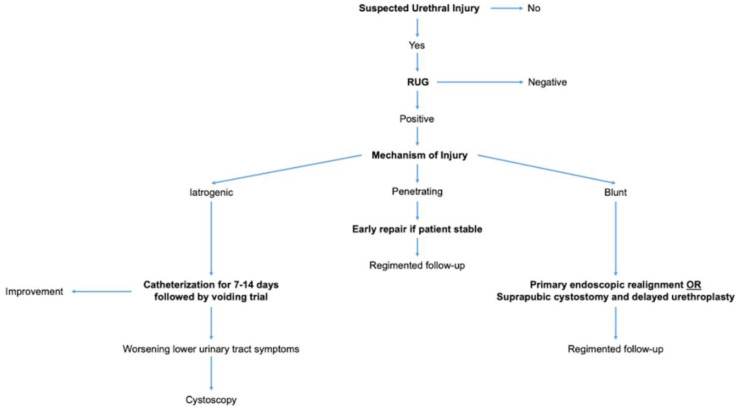
Diagnostic and Treatment Algorithm for Urethral Injury.

**Table 1 jcm-12-01495-t001:** Iatrogenic Urethral Trauma.

Author	Year	Level of Study	Number of Patients in Study	Diagnostic Procedures	Management	Complications
Kommu SS, Illahi I, Mumtaz F [16]	2007	4	NA	Endoscopy at the time of urological and gynecological procedure	NA	NA
Rosenstein DI, Alsikafi NF [6]	2006	4	NA	RUG	Non-surgical: Using catheters made of less irritative material, intermittent catheterization, transurethral microwave therapy Surgical: Urethrotomies and repeated dilations	Procedure: Urine extravasation and scarring
Maheshwari PN, Shah HN [40]	2005	4	7	RUG, Urethroscopy	Retrograde or suprapubic immediate endoscopic realignment	Procedure: Recurrence of stricture
Kashefi C, Messer K, Barden R, et al. [22]	2008	3	14	Clinical exam: urethral and/or perineal pain, blood at the meatus, non-draining catheter with ineffective irrigation Cystoscopy, RUG	Nursing education program	Injury: urethral scarring and strictures
Davis NF, Quinlan MR, Bhatt NR, et al. [31]	2016	3	37	Antegrade/RUG	One gentle catheter attempt for junior health professionals, Education, Percutaneous Insertion of SPC, Flexible Cystoscopy + guidewire, Transurethral catheter, Catheter manipulation, Rigid cystoscopy + guidewire, Open cystostomy + SPC	Procedural: Acute urinary retention, urosepsis, bleeding, Acute kidney injury, urethral stricture disease
Elliot S, McAninch JW, Chi T, et al. [24]	2006	3	48	NA	Anastomotic urethroplasty, flap urethroplasty, perineal urethrostomy, urethral stent	Injury: Urethral stenosis, rectourinary fistula

RUG = retrograde urethrogram, SPC = suprapubic catheter.

**Table 2 jcm-12-01495-t002:** Penetrating Anterior Urethral Trauma.

Author	Year	Level of Study	Number of Patients in Study	Diagnostic Procedures	Management	Complications
Dobrowolski ZF, Weglarz W, Jakubik P, et al. [44]	2002	4	255	RUG voiding cystourethrogram, endoscopy	Urethral repair	NA
Bjurlin MA, Kim DY, Zhao LC, et al. [36]	2013	4	162	RUG	Immediate primary urethral repair, urinary diversion with delayed reconstruction	Procedure: Urethral stricture
Bryk DJ, Zhao LC [14]	2016	4	NA	Clinical exam: rectal exam, blood at meatus, perineal or penile hematoma, voiding dysfunction RUG	Immediate open surgical repair, except when there are other life-threatening surgeries	NA
Kommu SS, Illahi I, Mumtaz F [16]	2007	4	NA	RUG under fluoroscopic guidance, anteroposterior radiograph	Mild contusions: conservative management SPC with delayed primary repair for severe contusions Partial or complete disruption: suprapubic diversion, primary realignment Exploration and debridement of devitalized tissue	Procedure: Urethral stricture Injury: Infection and necrotizing fasciitis
Cinman NM, McAninch JW, Porten SP, et al. [42]	2013	3	50	Clinical exam: gross hematuria	Initial SPC with delayed reconstruction, primary urethral repair with concomitant bladder neck or rectal injury	Procedure: Urethral stricture associated with urinary diversion

RUG = retrograde urethrogram, SPC = suprapubic catheter.

**Table 3 jcm-12-01495-t003:** Penetrating Posterior Urethral Trauma.

Author	Year	Level of Study	Number of Patients in Study	Diagnostic Procedures	Management	Complications
Koraitim MM, Reda IS [13]	2007	4	21	Pre-operative MRI of pelvis to aid in selection of appropriate surgical approach	End-to-end urethral anastomosis	Injury: Retropubic space scarring and fibrosis, fistula, impotence
Tausch TJ, Cavalcanti AG, Soderdahl DW, et al. [43]	2007	3	19	RUG	Primary urethral repair with concomitant bladder neck or rectal injury, delayed reconstruction, endoscopic realignment	Erectile dysfunction, pelvic abscess, incontinence, stricture
Cinman NM, McAninch JW, Porten SP, et al. [42]	2013	3	2	Clinical exam: gross hematuria	Initial SPC with delayed reconstruction, primary endoscopic realignment	Procedure: Urethral stricture associated with urinary diversion

RUG = retrograde urethrogram, MRI = magnetic resonance imaging.

## Data Availability

Not applicable.

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
