# Peer review of "Urethral Injuries: Diagnostic and Management Strategies for Critical Care and Trauma Clinicians"

_jcm, 2023, doi:10.3390/jcm12041495_

Round 1

Reviewer 1 Report

This is a review article summarizing current practice and standard about urethral injuries. The authors attempt to summarize a wide variety of urological problems for better understanding by critical care and trauma clinicians. 

Major problems

1.    The title clearly states that targeted main readers of this review are non-urology physicians specialized in critical care and primary management of trauma. As such, this review should aim at leading these readers to standard consensus shared by reconstructive urologists. The lack of high-grade evidence may not be exaggerated for that purpose. For example, the results of clinical trial cited as Ref 67 by Moses et al. has partially presented in AUA meeting 2021, showing no advantage for primary alignment, and this manuscript may include more warning against overusage of endoscopic alignment. On the other hand, it should be noted that few institute are capable of immediate reconstruction of penetrating or blunt trauma, leaving suprapubic and delayed urethroplasty as safe and general standard care.

2.    The description in regard of iatrogenic urethral injury is also misleading and confusing, because the authors state in Introduction section (the sentence from P1-2) that only one percent of urethral injury is related to transurethral procedure, but later they included variety of urethral injuries (1stparagraph in Page 5). Indeed, urethral injury represents significant etiology for urethral stricture (see Palminteri et al. Urology 2012). These include post-prostate surgery or radiation-induced stricture, but they may not be termed as ‘urethral trauma’, but rather complication of these therapies. I am not positive for including that such conditions are unrelated to physicians involved in critical care and primary management of trauma. The authors may exclusively discuss about acute catheter-related internal urethral trauma, which should be known by those potential readers.

3.    The tables include not only case studies, but also narrative reviews, which are based on authors’ personal view with no description of concrete patient number. The narrative reviews should not be included in the Table, but rather woven in the main text. 

Minor points

4.    Tables, I recommend more consistent terminology. For example, non-urological physician may be confused to face multiple idioms meaning one notion, for example, urethral repair, urethral reconstruction, urethroplasty, and surgery. Similarly, ascending urethrography or RUG? Urinary diversion, SPT, SPC or suprapubic cystostomy? Stricture or stenosis? 

Reviewer 2 Report

The authors summarized the updates on urethral injury, including injury patterns and trauma management. The review is generally well-written, and it is a very interesting topic.

Here are some minor comments:

1. A brief summary table for all three types of urethral injury is recommended.

2. Table 1: some information was placed in wrong columns (Year; Level of study);

3. For tables, it is recommended to sort each row by either "Year" or “Level of of Study”.

4. For Areas of Future Discovery and Improvement: more discussion regarding the critical care of patients is needed.
